

# Identification and validation of key genes associated with pathogenesis and prognosis of gastric cancer

Ai Li, Yan Li, Yueyue Li, Mingming Zhang, Hong Zhang and Feixue Chen

Department of Gastroenterology, Qilu Hospital, Cheeloo College of Medicine, Shandong University, Jinan, Shandong, China

## ABSTRACT

**Background**. Gastric cancer (GC) is the fourth leading cause of cancer-related death worldwide. However, the precise mechanisms and specific biomarkers of GC have not been fully elucidated. We therefore sought to identify and validate the genes associated with GC.

**Methods**. RNA sequencing was performed on gastric tissue specimens from 10 cases each of non-atrophic gastritis (NAG), intestinal metaplasia (IM), and GC. Validation of gene expression was conducted through immunohistochemistry (IHC) staining. The Kaplan–Meier Plotter database was utilized to screen genes associated with prognosis, while protein–protein interaction analysis was conducted to identify hub genes.

**Results**. In GC-IM, the differentially expressed genes (DEGs) were predominantly enriched in pathways related to ECM-receptor interaction, focal adhesion, PI3K-Akt pathway, and pathways in cancer. Conversely, in IM-NAG, the DEGs were primarily enriched in pathways associated with fat digestion and absorption, pancreatic secretion, and retinol metabolism. IHC staining revealed elevated expression levels of KLK7 and KLK10 in GC. Specifically, KLK7 expression was found to be correlated with differentiation ($P = 0.025$) and depth of invasion ($P = 0.007$) in GC, while both KLK7 and KLK10 were associated with the overall survival ($P < 0.05$). Furthermore, a total of ten hub genes from DEGs in GC-NAG (COL6A2, COL1A1, COL4A1, COL1A2, SPARC, COL4A2, FN1, PCOLCE, SERPINH1, LAMB1) and five hub genes in IM-NAG (SI, DPP4, CLCA1, MEP1A, OLFM4) were demonstrated to have a significant correlation with the prognosis of GC.

**Conclusions**. The present study successfully identified and validated crucial genes associated with GC, providing valuable insights into the underlying mechanisms of this disease. The findings of this study have the potential to inform clinical practice.

Corresponding author
Feixue Chen,
chenfeixue@email.sdu.edu.cn

Subjects Bioinformatics, Molecular Biology, Gastroenterology and Hepatology, Oncology
Keywords Gastric cancer, Pathogenesis, Prognosis, KLK7, KLK10

## INTRODUCTION

Gastric cancer (GC) ranks fifth among the most commonly diagnosed cancer and is the fourth leading cause of cancer-related death worldwide (*Sung et al., 2021*). Early diagnosis plays a crucial role in improving patient outcomes, as evidenced by the 92.6% 5-year survival rate of among early GC patients who undergo curative endoscopic submucosal dissection (*Suzuki et al., 2016*). However, challenges exist in achieving early diagnosis

due to the lack of specific biomarkers, insensitivity of imaging, and atypical clinical and endoscopic manifestations. Currently, approaches for early detection of GC are limited, with regular endoscopic examination for high-risk individuals being the primary method. However, most methods, such as blood biomarkers and imaging, only detect advanced and incurable GC (*Necula et al., 2019*). Despite the promise of new developments, such as liquid biopsies, in the diagnosis, treatment, and prognosis of GC, there is still a long way to go before their clinical application (*Tsujiura et al., 2014*). Therefore, it is imperative to seek novel and effective molecular biomarkers.

According to Correa's cascade, GC develops through a process of "normal mucosa-non-atrophic gastritis (NAG)-atrophic gastritis-intestinal metaplasia (IM)-intraepithelial neoplasia-GC," which is initiated by *Helicobacter pylori* (*Correa, 1992*). It remains controversial whether IM is the point of no return to this process (*Liou et al., 2020*). To fully investigate the underlying mechanisms of the process, we focused on three key stages in Correa's cascade: NAG, IM, and GC in our study.

Kallikrein 7 (KLK7) and Kallikrein 10 (KLK10) belong to Kallikrein (KLK) family, whose members participate in a vast range of normal and pathological processes (*Borgono & Diamandis, 2004*). Accumulating evidence has indicated that the KLK family is dysregulated in diverse cancer types. Some studies have demonstrated that KLK10 may serve as a biomarker with prognostic values in GC. However, the relationship between KLK10 expression and clinicopathological variables seems to be inconsistent across different studies (*Jiao et al., 2013*; *Kolin et al., 2014*). Until now, studies on the relationship between KLK7 and GC have been limited.

In the current study, we investigated the differentially expressed genes (DEGs) of GC, validated the expression of two DEGs, KLK7 and KLK10, and explored the key genes associated with the pathogenesis and prognosis of GC. The flowchart of the study is shown in Fig. 1. Our results provide a better understanding of the mechanisms involved in GC pathogenesis and novel biomarkers for diagnosis and prognosis of GC.

## MATERIALS AND METHODS

### Patients

Patients who underwent gastrectomy or gastroscopy from January 2019 to November 2020 in Qilu Hospital, Shandong University were enrolled. Patients undergone preoperative chemotherapy or radiotherapy, complicated with other primary tumors, or incapable to provide informed consent were excluded. Pathological staging of the GC patients was determined according to the 8th American Joint Committee on Cancer (AJCC) GC staging system. Pathological diagnosis was re-evaluated for all samples. The basic information was collected for analysis. The study was approved by the Institutional Ethics Committee of Qilu Hospital (approval number: 2018030). Before specimen collection, written informed consent was acquired from all patients.

### Specimens

Gastric tissues were obtained from patients. Tissues for RNA sequencing were stored in RNAlater (Invitrogen, Waltham, MA, USA) at 4 °C overnight and transferred to −80 °C for

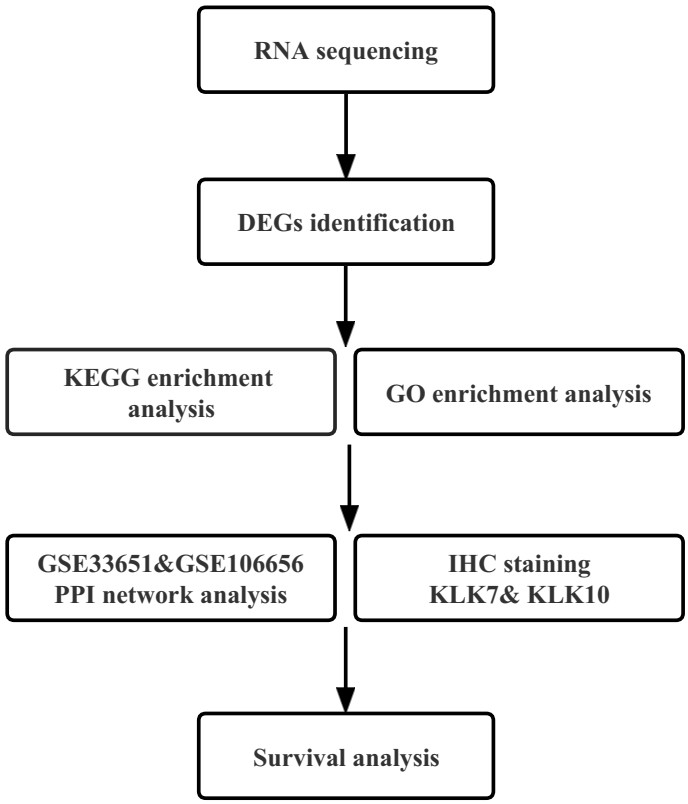

**Figure 1** **Workflow of the present study.** DEG, differentially expressed genes; GEO, Gene Expression Omnibus; KEGG, Kyoto Encyclopedia of Genes and Genomes; PPI, protein–protein interaction; IHC, immunohistochemistry.

storage. Tissues for immunohistochemistry (IHC) were fixed in 10% formalin overnight at 4 °C and then embedded in paraffin.

## RNA sequencing

Total RNA was isolated and purified using TRIzol reagent (Invitrogen, Waltham, MA, USA). Then, the poly(A) RNA was fragmented into small pieces and reverse-transcribed to create the cDNA. The processed cDNA products are amplified by PCR. Finally, RNA sequencing was performed on an Illumina Novaseq™ 6000 (LC-Bio Technology CO., Ltd., Hangzhou, China). RNA sequencing data is deposited in GSE191275.

## Identification of DEGs

Data were analyzed as previously described (*Pei et al., 2022*). The R software (4.1.0) and R package edgeR (version 3.14.0) were used to identify the DEGs between different groups (GC-NAG, IM-NAG, GC-IM). DEGs were defined with fold change (FC) > 2 or FC < 0.5 and $P < 0.05$.

## Gene ontology (GO) and kyoto encyclopedia of genes and genomes (KEGG) enrichment analysis

GO and KEGG enrichment analysis was conducted by R package "clusterProfiler." Three GO domains were included in GO enrichment analysis: biological process (BP), cellular component (CC), and molecular function (MF).

## Protein–protein interaction (PPI) network analysis

In this step, we included two other mRNA express files from GEO database (https://www.ncbi.nlm.nih.gov/geo/), GSE33651 and GSE106656. The overlapping genes of DEGs in GSE33651 and GC-NAG, GSE106656 and IM-NAG were analyzed by R package "VennDiagram." The PPI network analysis of these overlapping genes was visualized by the STRING database (https://string-db.org). Hub gene screening was performed using the CytoHubba plug-in of Cytoscape software.

## IHC staining

Briefly, 4 μm thick sections of the paraffin-embedded tissues were obtained and dried at 65 °C for 30 min. After deparaffinized by xylene solution and rehydrated in graded alcohols, sections were placed in sodium citrate buffer for 15 min at 95 °C to retrieve the antigen. The sections were rinsed three times with PBS and endogenous peroxidase blocking buffer was added for 15 min to block the endogenous peroxidase activity. Then, the sections were blocked with blocking solution for 30 min. Subsequently, the sections were incubated with primary anti-KLK7 antibody (1:300; PA5-27252, Invitrogen) and anti-KLK10 (1:500; ab229690, Abcam) overnight at 4 °C. After incubation with secondary antibody for 30 min and horseradish peroxidase for 20 min, the slides were stained with diaminobenzidine and hematoxylin. Two experienced pathologists blinded to the histopathologic features and patients' information observed the staining results independently using a light microscope.

The IHC score was assessed as previously described (*Fu et al., 2019*). One hundred cells in five high-power fields (×400) were observed. The staining intensity of positive cells was scored as 0 (negative), 1 (weak), 2 (moderate), and 3 (strong). The percentage of positively stained cells was scored as 0 (0%), 1 (1–25%), 2 (26–50%), 3 (51–75%), and 4 (76–100%). Multiplying these two scores got the total IHC score, which ranged from 0 to 12. High expression was defined as the total IHC score $\geq 4$, and low expression was defined as the total IHC score $< 4$.

## Prognostic analysis

Overall survival (OS) analysis and progression-free survival (PFS) were conducted on the Kaplan–Meier Plotter database (https://kmplot.com/analysis/index.php?p=service{&}cancer=gastric) (*Szasz et al., 2016*) to explore the prognostic values of selected genes.

## Statistical analysis

Statistical analysis was performed using SPSS 25.0 software. Comparison of clinical characteristics between the high and low expression group was determined by Chi-squared test or Fisher exact test. Comparison OS and PFS between groups was determined by Log-rank test. $P < 0.05$ obtained from a two-tailed test was considered statistically significant.

**Table 1 The demographic features of the patients involved in RNA sequencing and immunohistochemistry staining.**

| Patients | RNA-sequencing | | | Immunohistochemistry staining | | | | |
|---|---|---|---|---|---|---|---|---|
| | NAG | IM | GC | NAG | CAG | IM | EGC | GC |
| N | 10 | 10 | 10 | 17 | 15 | 13 | 16 | 13 |
| Gender (male/female) | 3/7 | 6/4 | 7/3 | 6/11 | 6/9 | 10/3 | 9/7 | 8/5 |
| Age (yr, mean ± SD) | 43.1 ± 11.2 | 53.7 ± 8.1 | 61.5 ± 11.5 | 47.9 ± 12.7 | 52.3 ± 11.3 | 53.2 ± 8.8 | 53.4 ± 4.2 | 67.6 ± 6.5 |

Notes.
NAG, non-atrophic gastritis; CAG, chronic atrophic gastritis; IM, intestinal metaplasia; EGC, early gastric cancer; GC, gastric cancer.

## RESULTS

### Identification of DEGs

Specimens of 30 patients (10 NAG, 10 IM, and 10 GC) were analyzed. The basic clinical characteristics of the patients were listed in Table 1. Hierarchical clustering analysis showed that the expression pattern of the GC group was significantly different from IM and NAG group (Fig. 2A). The specific number of identified DEGs between groups (NAG, IM, GC) was shown in Fig. 2B. As illustrated in the figure, the number of DEGs in IM-NAG ("IM-NAG" represents the comparison between the IM group and NAG group) is far less than DEGs in GC-IM or GC-NAG, which indicated that mRNA expression profiling of gastric tissue in GC patients was rather different from IM and NAG patients. In total, 1,257 (852 up-regulated and 405 down-regulated), 6,103 (4,639 up-regulated and 1,464 down-regulated), and 6,242 (4,773 up-regulated and 1,469 down-regulated) DEGs were identified separately in IM-NAG, GC-IM, GC-NAG. The top 10 up-regulated DEGs and down-regulated DEGs of GC-NAG and IM-NAG are presented in Table 2. Except for the presented DEGs, there are other genes in the list of the DEGs, such as KLK7, KLK10, COL4A1, TGM2, PDGFRB, etc. in GC-NAG; and CDX2, MUC2, TMEM139, TRIM36, SLC7A9, etc. in GC-IM.

### GO and KEGG enrichment analysis of DEGs

To fully investigate the biological roles of the DEGs, we chose DEGs in GC-NAG and IM-NAG to perform further analysis. GO enrichment analysis showed that DEGs in GC-NAG were enriched in signal transduction, multicellular organism development and cell differentiation in the BP, membrane, cytoplasm, and nucleus in CC and protein binding, metal-iron binding, and hydrolase activity in MF (Fig. 2C). KEGG enrichment analysis revealed that ECM-receptor interaction, focal adhesion, PI3K-Akt pathway, and pathways in cancer were mainly enriched pathways (Fig. 2D). Although DEGs in IM-NAG presented similar enriched BP compared with GC-NAG (Fig. 2E), KEGG enrichment analysis showed rather different results. Unlike GC-NAG, DEGs in IM-NAG were enriched in pathways like fat digestion and absorption, pancreatic secretion, neuroactive ligand–receptor interaction, and retinol metabolism (Fig. 2F).

### The expression and prognostic value of KLK7 and KLK10 in GC

In a previous study, through single-cell mRNA sequencing, KLK7 and KLK10 were selected as the signature specifically marking early GC (EGC) cells (*Zhang et al., 2020*).

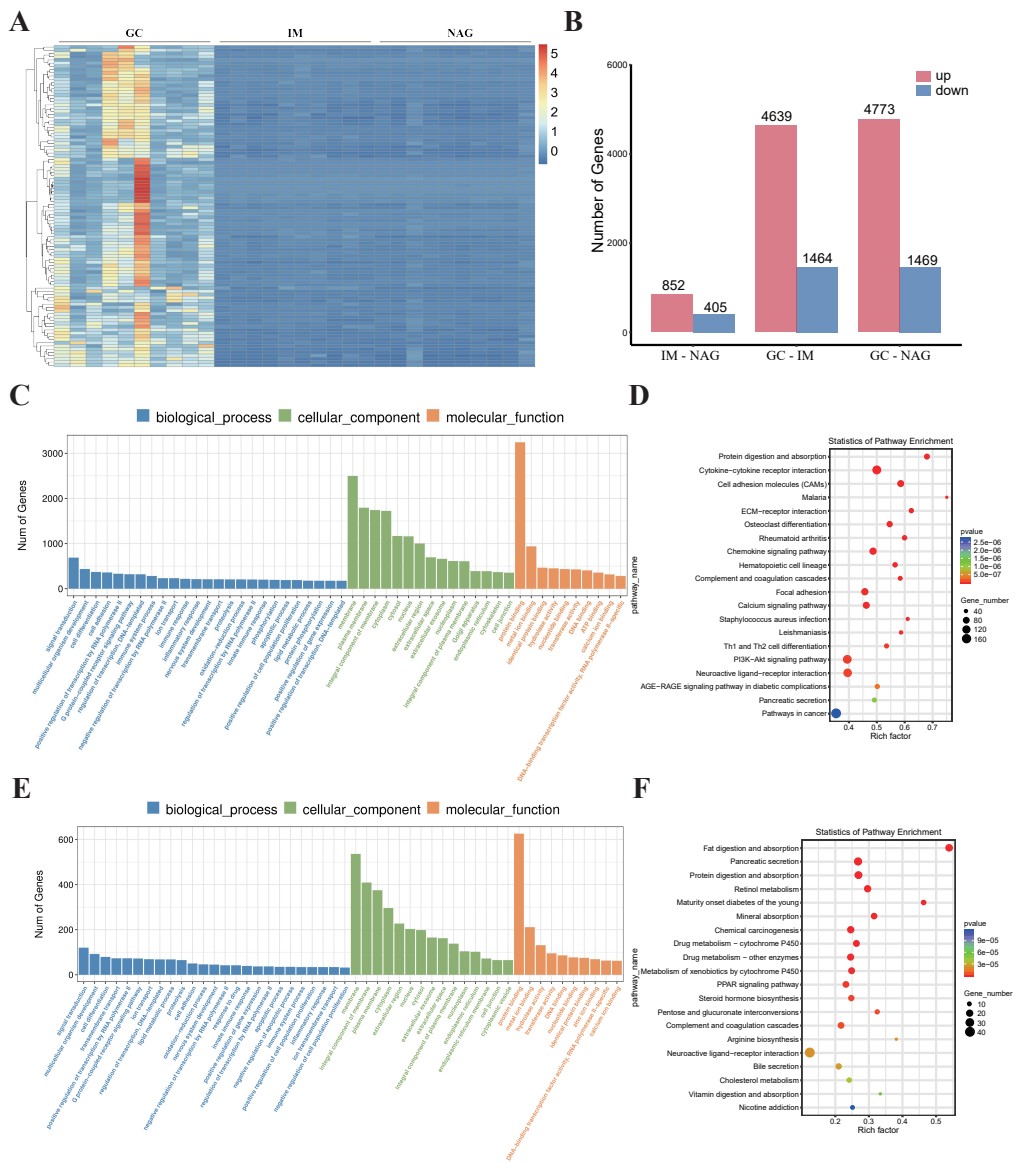

**Figure 2  The overall analysis of the DEGs.** (A) Heatmap of the DEGs. (B) The number of DEGs between different groups. (C) GO enrichment analysis of the DEGs in GC-NAG. (D) KEGG pathway enrichment analysis of the DEGs in GC-NAG. (E) GO enrichment analysis of the DEGs in IM-NAG. (F) KEGG pathway enrichment analysis of the DEGs in IM-NAG. DEGs, differentially expressed genes.

Interestingly, KLK7 and KLK10 were also in our DEGs in GC-NAG, so we further analyzed their expression in GC and gastric premalignant lesions. A total of 74 patients were included in this part of the study and they were divided into two groups: the non-cancer group (17 NAG, 15 CAG, and 13 IM) and the cancer group (16 EGC and 13 advanced GC (AGC)). The demographic features of the patients were shown in Table 1. IHC staining results showed that KLK7 was mainly expressed in the membrane of the epithelium while KLK10 was predominantly expressed in the cytoplasm of the epithelium. Both KLK7 and KLK10

**Table 2  The top ten upregulated DEGs and downregulated DEGs of GC-NAG and IM-NAG.**

| GC-NAG | | | | IM-NAG | | | | Regulation |
|---|---|---|---|---|---|---|---|---|
| Gene | log$_2$(FC) | P | P.adjust | Gene | log$_2$(FC) | P | P.adjust | |
| **Top ten up-regulated DEGs** | | | | | | | | |
| SERPINE1 | 6.07 | 3.52E−38 | 7.27E−34 | OTOP3 | 6.95 | 2.37E−32 | 4.48E−28 | Up |
| ADAMTS4 | 6.63 | 7.12E−36 | 7.35E−32 | HEPH | 4.63 | 6.55E−32 | 6.18E−28 | Up |
| WNT2 | 14.35 | 4.27E−29 | 2.94E−25 | SLC39A5 | 4.86 | 5.87E−26 | 3.69E−22 | Up |
| ATP1B3 | 3.1 | 6.44E−28 | 3.32E−24 | HOXA13 | 14.06 | 6.73E−25 | 3.18E−21 | Up |
| CCN1 | 6.2 | 1.39E−27 | 5.72E−24 | CDH17 | 6.83 | 3.49E−24 | 1.32E−20 | Up |
| MARCO | 9.28 | 3.50E−27 | 1.13E−23 | TINAG | 14.63 | 8.47E−24 | 2.37E−20 | Up |
| COL4A1 | 4.31 | 3.82E−27 | 1.13E−23 | ONECUT2 | 5.07 | 8.78E−24 | 2.37E−20 | Up |
| HEPH | 3.94 | 1.00E−26 | 2.58E−23 | MYO7B | 6.80 | 1.56E−22 | 3.68E−19 | Up |
| COL6A3 | 4.26 | 4.82E−26 | 1.04E−22 | CPS1 | 7.40 | 2.80E−22 | 5.88E−19 | Up |
| RGS1 | 5.42 | 5.04E−26 | 1.04E−22 | ADH6 | 5.08 | 3.97E−22 | 7.50E−19 | Up |
| **Top ten down-regulated DEGs** | | | | | | | | |
| ABCC5 | −2.09 | 2.65E−24 | 2.28E−21 | AMY2A | −13.77 | 6.75E−07 | 4.04E−05 | Down |
| RPTN | −9.5 | 6.62E−24 | 5.46E−21 | CTRC | −6.76 | 2.02E−06 | 1.08E−04 | Down |
| SLC7A8 | −2.78 | 2.07E−23 | 1.58E−20 | SYCP2 | −1.61 | 6.26E−06 | 2.88E−04 | Down |
| FLG2 | −11.9 | 1.69E−22 | 9.68E−20 | CASR | −1.73 | 6.55E−06 | 2.99E−04 | Down |
| SYTL2 | −2.64 | 3.97E−22 | 2.15E−19 | CYP2AB1P | −1.52 | 9.28E−06 | 4.09E−04 | Down |
| CHAD | −3.48 | 2.60E−21 | 1.28E−18 | SUCNR1 | −1.85 | 1.26E−05 | 5.32E−04 | Down |
| ALDOC | −3.19 | 2.14E−20 | 9.01E−18 | SYCN | −13.53 | 1.36E−05 | 5.68E−04 | Down |
| SMIM14 | −1.79 | 5.69E−20 | 2.13E−17 | CAMK2B | −1.71 | 1.37E−05 | 5.68E−04 | Down |
| SGSM3 | −1.89 | 3.11E−19 | 1.05E−16 | PAX6 | −1.67 | 1.87E−05 | 5.68E−04 | Down |
| RNASE4 | −2.07 | 4.64E−19 | 1.49E−16 | CPLX2 | −2.24 | 1.90E−05 | 7.44E−04 | Down |

**Notes.**

DEGs, differentially expressed genes; FC, fold change.

expressions were significantly higher in the cancer group than those of the non-cancer group ($P < 0.05$). Results are summarized in Table 3. Representative images are shown in Fig. 3A. KLK7 expression was significantly higher in patients with poor differentiation ($P = 0.025$) and advanced depth of invasion ($P = 0.007$). However, there was no significant correlation between KLK10 expression and clinicopathologic characteristics ($P > 0.05$) (Table 4). After confirming the expression of KLK7 and KLK10 in GC, we resort to the Kaplan–Meier plotter database to conduct the prognosis analysis. As shown in Figs. 3B and 3C, the high expression of KLK7 and KLK10 was significantly related to the poor OS of GC patients ($P < 0.05$). Besides, high expression of KLK10 (Fig. 3E), not KLK7 (Fig. 3D), showed negative correlation to PFS ($P < 0.05$).

## PPI network analysis of DEGs

To increase the credibility, two other datasets from the GEO database, GSE33651 and GSE106656, were involved in this process. GSE33651 is the expression profile of 40 GC tissue samples and 12 normal gastric tissue samples. GSE106656 is the expression profile of 7 IM tissue samples and 14 gastritis tissue samples. PPI analysis was performed after acquiring overlapping genes with the corresponding database separately (Figs. 4A, 5A).

**Table 3   KLK7 and KLK10 expression in gastric tissue.**

| Group | Type | KLK7 | | | | KLK10 | | | |
|---|---|---|---|---|---|---|---|---|---|
| | | N | High | Low | P | N | High | Low | P |
| Non-cancer | NAG | 17 | 0 | 17 | <0.001 | 17 | 2 | 15 | 0.005 |
| | CAG | 13 | 2 | 11 | | 15 | 5 | 10 | |
| | IM | 13 | 3 | 10 | | 12 | 2 | 10 | |
| Cancer | EGC | 15 | 8 | 7 | | 16 | 7 | 9 | |
| | AGC | 13 | 12 | 1 | | 13 | 8 | 5 | |

Notes.

NAG, non-atrophic gastritis; CAG, chronic atrophic gastritis; IM, intestinal metaplasia; EGC, early gastric cancer; AGC, advanced gastric cancer.

The hub genes of DEGs in GC-NAG included COL6A2, COL1A1, COL4A1, COL1A2, SPARC, COL4A2, FN1, PCOLCE, SERPINH1, and LAMB1 (Fig. 4B). These genes were enriched in the ECM-receptor interaction pathway and all associated with the prognosis of GC (Fig. 4C). The hub genes of DEGs in IM-NAG included SI, DEFA6, DEFA5, DPP4, CLCA1, MEP1A, MEP1B, REG4, OLFM4, SLC5A1 (Fig. 5B). These genes were enriched in protein digestion and absorption and carbohydrate digestion and absorption pathway. Further analysis revealed that six genes (SI, DPP4, CLCA1, MEP1A, REG4, OLFM4) were in the list of DEGs in GC-IM. Besides, apart from REG4, another five genes were associated with the prognosis of GC (Fig. 5C).

## DISCUSSION

In our study, we analyzed the gene expression profiles of GC, IM, and NAG through RNA sequencing. The overall analysis of DEGs revealed that the expression profiles of the GC group were different from the IM group and NAG group. The KEGG pathway enrichment analysis also revealed significant differences in enriched pathways of DEGs in GC-NAG and IM-NAG. In GC-NAG, the majority of DEGs were enriched in pathways such as ECM-receptor interaction, focal adhesion, PI3K-Akt pathway, and pathways in cancer. On the other hand, DEGs in NAG-IM were primarily associated with pathways involved in fat digestion and absorption, pancreatic secretion, neuroactive ligand–receptor interaction, and retinol metabolism. These findings lay a solid foundation for investigating the underlying biological processes of gastric cancer.

The KLK family consists of 15 homologous secreted trypsin or chymotrypsin-like serine proteases (*Borgono & Diamandis, 2004*). Numerous studies have indicated dysregulation of kallikrein expression in various cancer types, which is associated with prognosis (*Dorn et al., 2011*; *Hua et al., 2021*; *Lilja, Ulmert & Vickers, 2008*; *Obiezu et al., 2001*; *Zhu et al., 2018*). Several members have been shown to be related to GC. For instance, KLK6 exhibits significant overexpression in GC and holds potential as a robust prognostic indicator (*Nagahara et al., 2005*). However, further exploration is required for other members.

Through RNA sequencing, we observed a significant upregulation of KLK7 and KLK10 expression in GC compared with NAG, and this finding was was further confirmed by IHC staining. High expression of these genes indicated a poor prognosis of GC. Moreover,

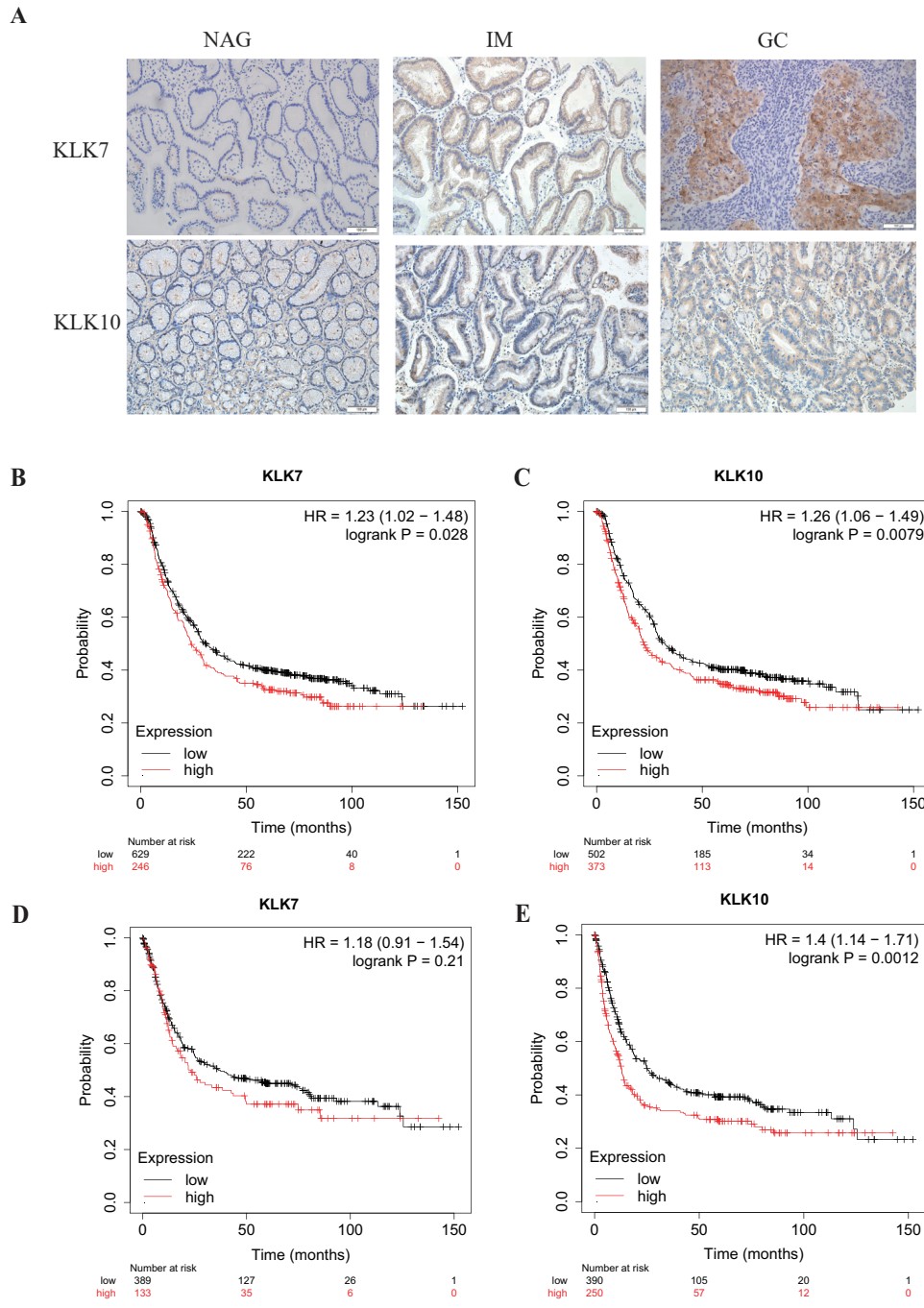

**Figure 3** **Verification of KLK7 and KLK10 expression in gastric cancer.** (A) Representative immuno-histochemical (IHC) staining of KLK7 and KLK10 in GC and NAG (×200). (B, C) OS analysis KLK7 and KLK10 in GC. (D, E) PFS analysis KLK7 and KLK10 in GC. KLK7, Kallikrein 7; KLK10, Kallikrein 10; NAG, non-atrophic gastritis; GC, gastric cancer. HR, Hazard Ratio.

**Table 4  The correlation between KLK7, KLK10 expression and clinicopathological characteristics.**

| Clinicopathologic characteristics | KLK7 (n = 28) | | | | KLK10 (n = 29) | | | |
|---|---|---|---|---|---|---|---|---|
| | N | High (%) | Low (%) | P | N | High (%) | Low (%) | P |
| Age (yrs) | | | | 1.000 | | | | 1.000 |
| <60 | 15 | 11 (73.3%) | 4 (26.7%) | | 16 | 8 (50%) | 8 (50%) | |
| ≥60 | 13 | 9 (69.2%) | 4 (30.8%) | | 13 | 7 (53.8%) | 6 (46.2%) | |
| Sex | | | | 0.083 | | | | 0.060 |
| Male | 16 | 10 (62.5%) | 6 (37.5%) | | 17 | 6 (35.3%) | 11 (64.7%) | |
| Female | 12 | 11 (91.7%) | 1 (8.3%) | | 12 | 9 (75%) | 3 (25%) | |
| Tumor size (cm) | | | | 0.462 | | | | 0.700 |
| <5 | 18 | 12 (66.7%) | 6 (33.3%) | | 19 | 9 (47.4%) | 10 (52.6%) | |
| ≥5 | 10 | 8 (80%) | 2(20%) | | 10 | 6 (60%) | 4 (40%) | |
| Differentiation status | | | | 0.025* | | | | 0.466 |
| Well + moderate | 15 | 8 (53.3%) | 7 (46.7%) | | 15 | 9 (60%) | 6 (40%) | |
| Poor | 13 | 12 (92.3%) | 1 (7.7%) | | 14 | 6 (42.9) | 8 (57.1%) | |
| Depth of invasion | | | | 0.007* | | | | 0.264 |
| T1 + T2 | 15 | 8 (53.3%) | 7 (46.7%) | | 17 | 7 (41.2%) | 10 (58.8%) | |
| T3 + T4 | 13 | 13(100%) | 0 (0%) | | 12 | 8 (66.7%) | 4 (33.3%) | |
| Lymph node metastasis | | | | 0.401 | | | | 1.000 |
| Positive | 12 | 10 (83.3%) | 2 (16.7%) | | 13 | 7 (53.8%) | 6 (46.2%) | |
| Negative | 16 | 10 (62.5%) | 6 (37.5%) | | 16 | 8 (50%) | 8 (50%) | |
| TNM stage | | | | 0.086 | | | | 0.651 |
| I + II | 22 | 14 (63.6%) | 8 (36.4%) | | 23 | 11 (47.8%) | 12 (52.2%) | |
| III + IV | 6 | 6 (100%) | 0 (0%) | | 6 | 4 (66.7%) | 2 (33.3%) | |

**Notes.**
*P < 0.05.

KLK7 was found to be associated with differentiation and depth of invasion in GC whereas KLK10 showed no correlation with clinicopathologic characteristics. A previous study has demonstrated that KLK7 expression is upregulated in GC cells under acidic environment, leading to enhanced cell invasion (*Lim et al., 2020*).

To our knowledge, no previous study has investigated the association between KLK7 expression and clinicopathological characteristics in GC. Thus, our study conducted a preliminary investigation in this area. We believe that KLK7 could serve as a potential therapeutic target for GC progression, although this hypothesis requires further validation. Several studies have demonstrated that KLK10 expression is upregulated in GC and elevated KLK10 expression predicts poor prognosis (*Feng et al., 2006*; *Jiao et al., 2013*), which is consistent with our result. Nonetheless, the results of the relationship between KLK10 expression and clinicopathological characteristics remains contentious. Although a previous study observed a positive relationship between KLK10 expression and lymph node metastasis as well as depth of invasion (*Jiao et al., 2013*), our study did not find any correlation between KLK10 expression and clinicopathological characteristics. Given the limited sample size in both studies, it is important to conduct further investigations to thoroughly examine their relationship.

Peer J

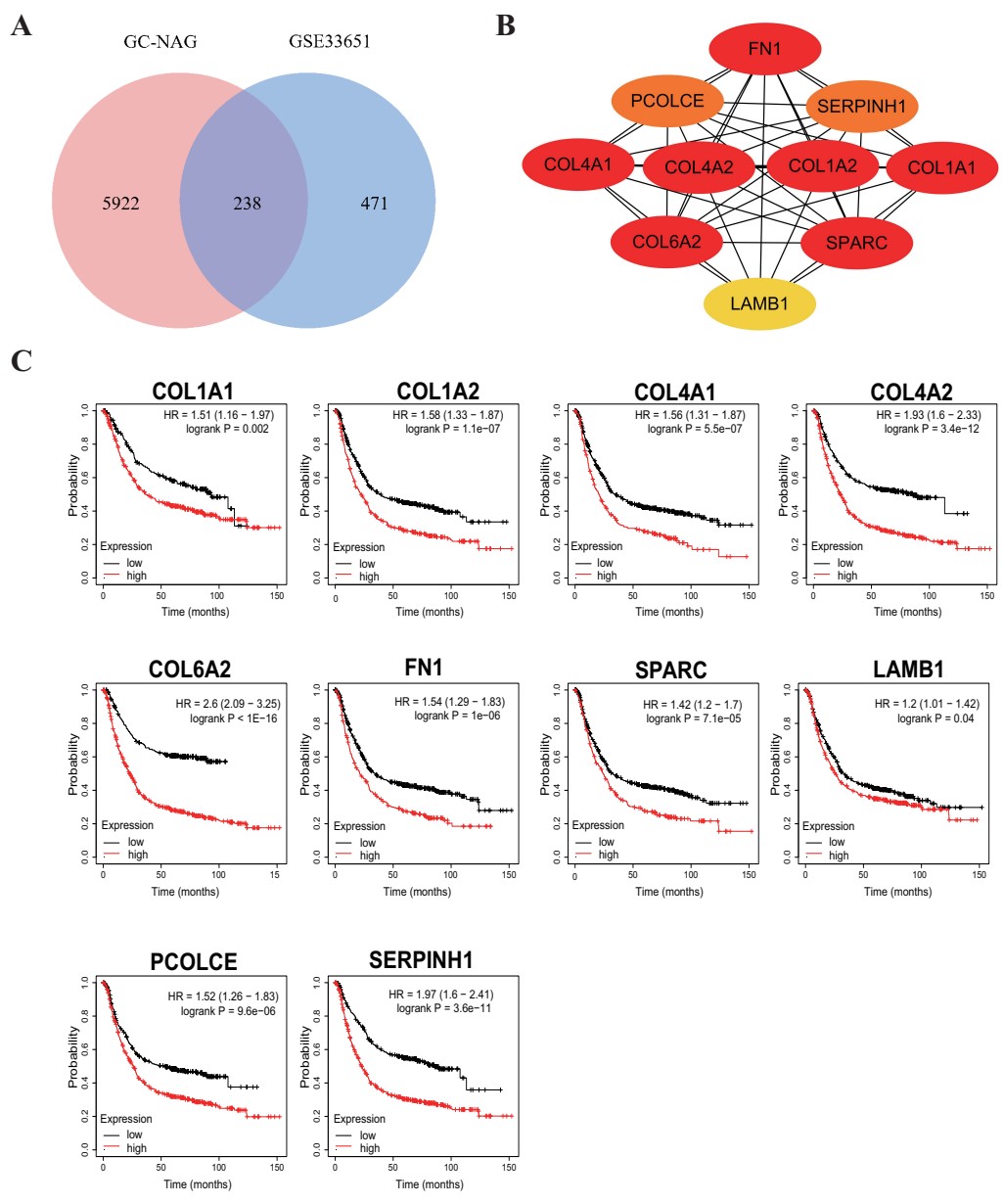

**Figure 4** **Hub genes in GC-NAG and the prognostic value in GC.** (A) Venn diagram showing the overlapping DEGs between our result and GSE33651. (B) PPI network diagram of the hub genes (genes with darker color rank higher). (C) Survival analysis of the hub genes. DEG, differentially expressed genes; PPI, protein-protein interaction.

In our subsequent analysis, we constructed a PPI network of DEGs and identified key genes within the network. The construction of the network will enhance our understanding of the interactions among DEGs. Subsequent analysis revealed that the selected hub genes in GC-NAG were all associated with the poor prognosis in GC. Interestingly, five genes (SI,

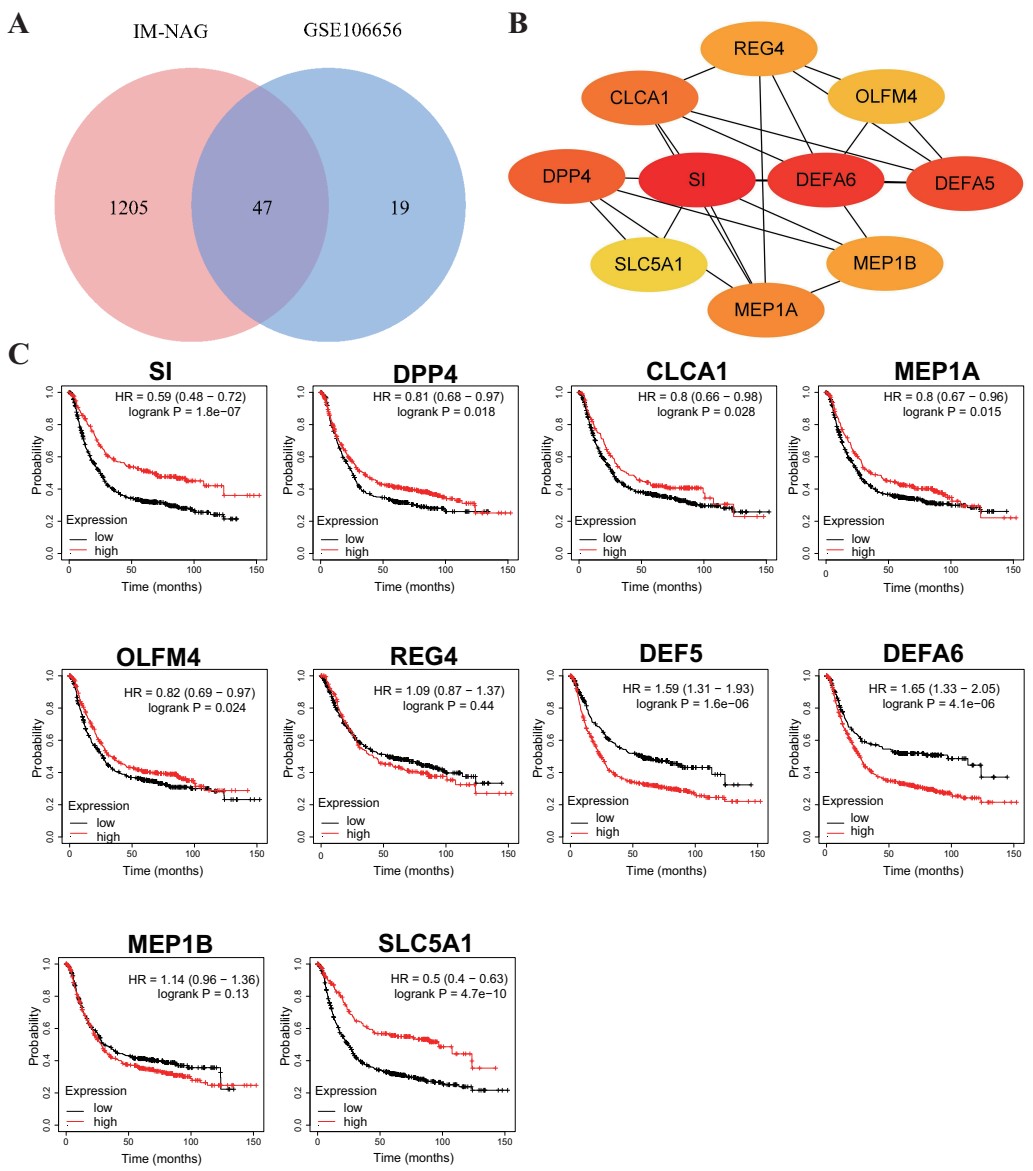

**Figure 5   Hub genes in IM-NAG and the prognostic value in GC.** (A) Venn diagram showing the over-lapping DEGs between our result and GSE106656. (B) PPI network diagram of the hub genes (genes with darker color rank higher). (C) Survival analysis of the hub genes. DEG, differentially expressed genes; PPI, protein-protein interaction.

DPP4, CLCA1, MEP1A, OLFM4) among the ten hub genes in IM-NAG were also present in the list of DEGs in GC-IM and showed association with the GC prognosis. This suggests that the expression of certain crucial genes in GC-IM has already undergone changes during the transition from NAG to IM, potentially exerting a significant impact on GC

pathogenesis. These findings improve our understanding of the underlying mechanisms of GC.

GC develops through a multistep process involving multiple genetic and epigenetic changes. The Operative Link for Gastric Intestinal Metaplasia assessment (OLGIM), is a risk classification system that evaluate GC risk by considering the severity and distribution of IM. High-risk OLGIM stages have demonstrated strong predictive value for GC, Highlighting the significance of investigating IM (*Yun et al., 2018*). Importantly, our findings revealed a significant increase in the expression of CDX2 in the IM group compared to NAG group, supporting its pivotal role in the development and maintenance of intestinal metaplasia (*Barros et al., 2012*). Overall, our study offers valuable insights for a better comprehension of GC development and contributes to the identification of novel biomarkers for GC diagnosis.

Among the ten hub genes in GC-NAG, five genes belong to the collagen family: COL1A1, COL1A2, COL4A1, COL6A2, COL4A2. These genes have already been identified as potential diagnostic and prognostic biomarkers, as well as possible therapeutic targets in GC (*Cao et al., 2018*; *Chen et al., 2020*; *Feng et al., 2020*; *Jiang et al., 2019*; *Li, Ding & Li, 2016*; *Wang et al., 2020*; *Zhao et al., 2021*). These genes are enriched in ECM-receptor interaction, focal adhesion, and PI3K-Akt signaling pathway, all of which have been proven to be closely related to the pathogenesis of various cancer types (*Alzahrani, 2019*; *Bao et al., 2019*; *Ediriweera, Tennekoon & Samarakoon, 2019*; *Eke & Cordes, 2015*; *Machackova et al., 2020*; *Paluch, Aspalter & Sixt, 2016*). Fibronectin 1 (FN1), an extracellular matrix protein, has been identified as a key gene in GC (*Zhao et al., 2021*). It plays a key role in inhibiting the proliferation, migration, and invasion of GC cells (*Zhang et al., 2017*). Secreted protein acidic and cysteine-rich (SPARC) also plays a key role in cancer through extracellular matrix remodeling and promoting epithelial-mesenchymal transition (*Camacho et al., 2020*). Previous studies have shown a close association between SPARC and the progression and poor survival of GC (*Li et al., 2019*; *Yin et al., 2010*; *Zhao et al., 2010*). A recent study suggested that Procollagen C-endopeptidase enhancer (PCOLCE) is a potential prognostic biomarker associated with immune infiltration in GC (*Xiang et al., 2020*). Serpin H1 (SERPINH1), a collagen-binding protein, has also been shown to be a hub gene with prognostic value in GC (*Li et al., 2018*). It is involved in regulating mesenchymal transition and GC metastasis (*Tian et al., 2020*). Laminin subunit beta-1 (LAMB1) is believed to be associated with T stage and poor prognosis in GC. Upregulation of LAMB1 could promote GC growth and motility (*Lee et al., 2021*; *Ran et al., 2021*).

Interestingly, five hub genes in IM-NAG were associated with the prognosis of GC: SI, DPP4, CLCA1, MEP1A, and Olfactomedin 4 (OLFM4). OLFM4 expression is thought to be involved in early gastric carcinogenesis and is of prognostic significance in advanced GC (*Jang, Lee & Kim, 2015*). Depletion of the OLFM4 gene inhibits cell growth and increases apoptosis in GC cells, indicating that OLFM4 is a potential target (*Liu et al., 2012*). Therefore, all the hub genes may play key roles in GC and have the potential to interact with each other. They can be considered as potential effective candidates for early diagnosis or prognosis. Further analysis of these genes will certainly contribute to

establishing a comprehensive understanding of the underlying mechanisms and identifying more molecular targets for the treatment of GC.

Certainly, our study had some limitations. First, the relatively small sample size may not be sufficient for clinical application. Second, further investigation is needed to fully understand the underlying mechanism of the selected genes. Finally, large-scale clinical studies are necessary before implementing these results into clinical practice. These questions require further study. While our study may have certain limitations, it has succeeded in revealing significant aspects that possess the potential to stimulate and guide future investigations. Diverging from other studies, our approach capitalizes on a comprehensive understanding of GC's multistage evolution, thus enhancing the opportunity to unearth novel insights into its pathogenesis and pioneer innovative treatment modalities.

## CONCLUSIONS

In conclusion, we have identified hub genes and key pathways associated with GC, and we have also validated the expression of KLK7 and KLK10 in GC. Our study holds the promise of unearthing novel insights into the pathogenesis of GC and fostering innovative therapeutic strategies.

## ACKNOWLEDGEMENTS

We would like to express our gratitude to Professor Wenbin Yu and Cheng Chen for their valuable assistance in this study.

### Funding
The authors received no funding for this work.

### Competing Interests
The authors declare there are no competing interests.

### Author Contributions
- Ai Li performed the experiments, analyzed the data, authored or reviewed drafts of the article, and approved the final draft.
- Yan Li analyzed the data, prepared figures and/or tables, authored or reviewed drafts of the article, and approved the final draft.
- Yueyue Li analyzed the data, prepared figures and/or tables, and approved the final draft.
- Mingming Zhang analyzed the data, prepared figures and/or tables, and approved the final draft.
- Hong Zhang conceived and designed the experiments, prepared figures and/or tables, and approved the final draft.
- Feixue Chen conceived and designed the experiments, prepared figures and/or tables, and approved the final draft.

## Human Ethics

The following information was supplied relating to ethical approvals (i.e., approving body and any reference numbers):

The study was approved by the Institutional Ethics Committee of Qilu Hospital.

## Data Availability

Data is available at GEO: GSE191275 and the accession numbers for each specimen and their details have also been uploaded as a Supplemental File.

## Supplemental Information

Supplemental information for this article can be found online at http://dx.doi.org/10.7717/peerj.16243#supplemental-information.

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
