# Peer review of "Identification and validation of key genes associated with pathogenesis and prognosis of gastric cancer"

_PeerJ, doi:10.7717/peerj.16243_

## Round 0.1 · original submission · Minor Revisions

The author is requested to revise the manuscript according to the reviewer's comments.

Reviewer 1 ·

Basic reporting

no comment

Experimental design

no comment

Validity of the findings

no comment

Additional comments

1. The size of Figure 2 is small and hard to see. The characters in the graphs in Figures 4 and 5 are small and difficult to read.
2. No correlation was found between KLK10 expression and clinicopathological features. What are you thinking besides sample size? Have you performed qPCR or western blot for KLK7, KLK10, COL1A1, etc.?
3. The conclusion is suitably cautious about the findings. It would be helpful to discuss how these results compare with those of other similar studies. Also, more specific suggestions for future research and potential clinical applications could strengthen the conclusion.

·

Basic reporting

In the Introduction Part, there is still a long way to go before their clinical application. What’s the biggest obstacle for clinical application of gastric cancer (GC)?

Experimental design

1. Why only select patients undergoing gastrectomy or gastroscopy from January 2019 to November 2020? What’s the 8th American Joint Committee on Cancer (AJCC) GC staging system?
2. Why did the authors conduct RNA sequencing experiments with 10 biological replicates? How many technical replicates for RNA sequencing experiments?
3. The staining intensity of positive cells was scored as 0 (negative), 1 (weak), 2 (moderate), and 3 (strong). How to identify negative, moderate, strong from image?

Validity of the findings

What’s the representative function of COL1A1, COL1A2, COL4A1, COL6A2, COL4A2? Do there have any mechanism related with GC?

Additional comments

N/A

Reviewer 3 ·

Basic reporting

The manuscript delivers a good information regarding key genes associated with pathogenesis and prognosis of gastric cancer. The authors claim that the work successfully identified and validated crucial genes associated with GC and findings of this study have the potential to inform clinical practice.

The workflow of the project seems to be simple yet sturdy to have a deep information regarding GC related DEGs and survival analysis.

Experimental design

The technical aspect and approach seem to be sturdy though methodology section could have been better especially under heading of-
Identification of DEGs
Also I would highly recommend to include the versions of all the R/Bioconductor packages used in this work.

Validity of the findings

While the work seems to be promising I do have some concerns -

1.The nomenclature for the conditions seemed to be non-consistent/ less explanatory in nature. For example- The authors say that RNA sequencing was performed on gastric tissue specimens from 10 cases each of non-atrophic gastritis (NAG), intestinal metaplasia (IM), and GC and then in the results/figures they labeled - IM-NAG, GC-IM and GC-NAG. Though I understand the abbreviation, it is a bit confusing in order to compare the stages.

2. Fig 2A- labels are not legible at all. Also I would recommend authors to reproduce the plot and label/cluster the column too based on compared groups.

3. Fig 2 D and G- Labels too small.

4. Table 2- Is the p -value shown in the table is corrected ?

5. Maybe include a color gradient explanation in the plot itself elaborating PPI genes (fig 4 and 5B)

---

## Round 0.2 · accepted · Accept

All three reviewers agree to accept the manuscript. I also do not find this document to be a significant risk of publication, so I agree to accept it for publication.

Reviewer 1 ·

Basic reporting

no comment

Experimental design

no comment

Validity of the findings

no comment

Additional comments

no comment

·

Basic reporting

The manuscript have been modified according to the suggestion of reviewers.

Experimental design

The experimental design is acceptable for publication.

Validity of the findings

Identification and validation of key genes would be helpful for understanding the mechanism of pathogenesis and prognosis of gastric cancer

Reviewer 3 ·

Basic reporting

See the additional comments

Experimental design

See the additional comments

Validity of the findings

See the additional comments

Additional comments

The authors improved the manuscript by incorporating the responses to the points raised in the review. Fig2 a color legend is still not legible. Kindly fix that. Other than that I would suggest to accept the manuscript.